# Effects of the Mobility-Fit Physical Activity Program on Strength and Mobility in Older Adults in Assisted Living: A Feasibility Study

**DOI:** 10.3390/ijerph19095453

**Published:** 2022-04-29

**Authors:** Yijian Yang, Kimberley S. van Schooten, Vicki Komisar, Heather A. McKay, Joanie Sims-Gould, Debbie Cheong, Stephen N. Robinovitch

**Affiliations:** 1Department of Sports Science and Physical Education, The Chinese University of Hong Kong, Hong Kong, China; 2CUHK Jockey Club Institute of Ageing, The Chinese University of Hong Kong, Hong Kong, China; 3Neuroscience Research Australia, University of New South Wales, Sydney, NSW 2033, Australia; kim.vanschooten@gmail.com; 4School of Public Health and Community Medicine, University of New South Wales, Sydney, NSW 2033, Australia; 5School of Engineering, The University of British Columbia—Okanagan Campus, Kelowna, BC V1V 1V7, Canada; vicki.komisar@ubc.ca; 6Centre for Hip Health and Mobility, University of British Columbia, Vancouver, BC V5Z 1M9, Canada; heather.mckay@ubc.ca (H.A.M.); joanie@joaniesimsgould.com (J.S.-G.); stever@sfu.ca (S.N.R.); 7British Columbia Women’s Health Centre, Vancouver, BC V6H 3N1, Canada; debbie.cheong@cw.bc.ca; 8Department of Biomedical Physiology and Kinesiology, Faculty of Science, Simon Fraser University, Burnaby, BC V5A 1S6, Canada

**Keywords:** elderly, care facility, physical activity, mobility, upper limb strength, fall prevention

## Abstract

Physical activity programs focusing on fall prevention often overlook upper-limb strength, which is important for transferring, balance recovery, and arresting a fall. We developed and evaluated a physical activity program, Mobility-Fit for older adults in Assisted Living (AL) that includes upper-limb strengthening, agility, coordination, and balance exercises. Thirty participants (85 ± 6 years) were recruited from two AL facilities; 15 were assigned to Mobility-Fit (three times/week, 45 min/session for 12 weeks) and 15 maintained usual care. Twenty-two participants (11 in each group) completed the study. We compared outcome changes between groups and interviewed participants and staff to explore the effectiveness and feasibility of the program. Among participants who attended Mobility-Fit, knee extension strength increased by 6%, reaction time decreased by 16%, and five-time sit-to-stand duration decreased by 15%. Conversely, participants in the usual care group had a 6% decrease in handgrip strength. Changes of these outcomes were significantly different between two groups (*p* < 0.05). Participants enjoyed the program and staff suggested some changes to improve attendance. Our results indicate that Mobility-Fit is feasible to deliver and beneficial for older adults in AL and may guide future clinical trials to evaluate the effectiveness of upper limb strengthening on safe mobility of older adults in care facilities.

## 1. Introduction

The proportion of adults aged 65 and over will nearly double to 25% by 2036 in Canada [1]. With increasing life expectancy, an increased number of older adults will live with disability for longer, which prompts a subsequent rise in the demand for care. Assisted living (AL) bridges the gap between independent living and long-term care. AL is a housing option for older adults who need support with activities of daily living (ADLs), but do not require the full-time care provided to more frail seniors in costly long-term care environments. Reasons for subsequent placement of AL residents into long-term care include mobility impairments, falls, and increased dependence on others to perform ADLs [2]. Thus, there is an urgent need to maintain older adults’ mobility and decrease their risk of falls in AL, so as to prevent or delay their transition to higher levels of care.

Physical activity is an established strategy for preserving function and encouraging mobility across the lifespan. The benefit of physical activity may be particularly high among AL residents, of whom nearly 90% are sedentary [3,4]. When developing physical activity programs in AL, it is key to introduce challenging activities in ways that are safe and enjoyable [5], to improve the capacity of older adults to remain mobile, perform ADLs, and avoid fall-related injuries. Older adults in care facilities who experience declines in upper limb strength have decreased capacity to transfer between chairs or between sitting and standing and have difficulty in recovering balance via handrail grasping and using the upper limbs to arrest falls [6,7,8,9,10].

Therefore, we developed the Mobility-Fit program, adapted from Osteofit [11], to address these key issues. Mobility-Fit is designed to improve the mobility of older adults in AL, with a focus on enhancing agility and upper-limb and core strength. Mobility-Fit for AL residents builds upon the design of “Osteofit” [11] and “Staying on Your Feet” interventions [12] for community-dwelling older adults. Osteofit significantly enhanced lower limb strength and balance [11], and the Staying on Your Feet program improved upper limb strength and fall risk [12]. Our Mobility-Fit complements fall prevention strategies that aim to preserve or improve the capacity of older adults to maintain balance, recover from balance loss, and arrest falls when they occur to avoid subsequent disabling injury (e.g., traumatic brain injury) [8,10].

The present study addresses two key objectives: (1) to describe the development of Mobility-Fit and (2) to explore the feasibility of Mobility-Fit delivered to older adults who reside in AL. Our mixed-method study integrates quantitative (strength and mobility measures) and qualitative (participant feedback) outcomes. We hypothesized that strength and mobility would be preserved in AL residents who attended Mobility-Fit, compared to older adults who received usual care.

## 2. Materials and Methods

### 2.1. Ethics Statement

Research ethics boards at the University of British Columbia, Simon Fraser University, and Fraser Health Authority approved this study. All participants provided written informed consent before participating in this study.

### 2.2. Study Design and Recruitment

We conducted this quasi-experimental, two-group, and mixed (quantitative and qualitative) methods study between January 2017 and October 2017. We recruited 30 participants through poster advertisements and staff referrals in two AL facilities affiliated with the Fraser Health Authority, British Columbia (Maple Ridge, BC, Canada). The ELIM Village had 109 apartments and Swedish Canadians (SC) Assisted Living Residence had 64 apartments, housing older adults who need assistance in ADL. We conducted an information session on-site for interested people who wished to learn more about the study. Our inclusion criteria were: (1) 65 years or older, (2) able to express personal preferences clearly, (3) able to understand and follow instructions in English, and (4) able to get up from sitting to standing and walk for 10 m with or without a walking aid, determined by self-report and consultation with care staff. Exclusion criteria were: (1) bedridden or confined in wheelchair, (2) physically disabled, (3) having agitated behaviour. Our team fully described the study to AL residents; those who were interested in participating provided written consent.

### 2.3. Sample Size Consideration

Feasibility studies generally do not require a formal sample size calculation [13]. We decided to recruit a sample of 30 residents who are diverse in age, cognitive function, and functional capacity to gain insight into the feasibility of a future large-scale study, which is aligned with recommendations for pilot studies [14,15]. We included a control group to explore the potential effect of time on main outcomes, given that there was no a priori data available of decline in upper limb and core strength in frail older adults in assisted living in Canada. Data from this feasibility study will allow us to obtain effect size estimates to power a full-scale randomized controlled trial for which we are seeking funding.

### 2.4. Participant Recruitment and Group Allocation

Twenty older adults from ELIM (5 men, 15 women) and 10 from the SC residence (5 men, 5 women) (Figure 1) were recruited and non-randomly allocated to two groups. Specifically, the first 15 participants from ELIM were assigned to the Mobility-Fit intervention group. The last five participants from ELIM and all 10 participants from SC were assigned to the Usual Care group. The Usual Care group received the facility’s routine daily activities (mainly stretching movements of upper and lower limbs). Usual care routines were similar between the two AL facilities. We did not use randomization as our main purpose was to examine the feasibility of the Mobility-Fit program. Furthermore, resident characteristics between AL facilities of Fraser Health Authority are similar based on their admission criteria. As shown in Table 1, participant demographics (*n* = 30) were similar between the two groups, except for the body mass index. Two trained research assistants conducted cognitive and physical assessments at baseline (T0) and at week 12 (completion of the study; T1). The research assistants communicated weekly with all participants during the study. These contacts occurred between measurement sessions to encourage both groups to remain involved in the study.

### 2.5. Development and Delivery of Mobility-Fit

We collaborated with the Provincial Osteofit Coordinator and physiotherapists at Fraser Health Authority to develop Mobility-Fit; some elements were adapted from the Osteofit program [11]. Mobility-Fit retained key components of Osteofit that were previously shown to be effective in community-dwelling older adults for enhancing lower limb function and reducing falls risk [11], while adding new components designed to enhance upper limb (wrist, triceps, shoulder) and core strength.

In brief, Mobility-Fit is a 12-week program, with three 45-min sessions per week. The program was delivered at ELIM care facility. A session begins with a 5-min warm up that includes joint range of motion (ROM) in all planes of movement. After the warm-up, participants begin exercises while seated and then progress to exercises while standing. The 40-min main workout consists of: (1) postural strength, with frequent lumbar extension and shoulder external rotation; (2) balance and core strength, with sitting away from the back of the chair and mobilizing arms while maintaining trunk stability, and gradually progressing to standing and decreased base of support (using knee lifts); (3) agility (reaction and speed), including ball toss, random foot placements, coordination, and multi-tasking activities; (4) lower limb strength, including hip abduction, hip extension, heel raises, and hamstring curls while standing behind and holding onto a chair; (5) wrist strength, with wrist curls (pronated/supinated, in rotation, and using bean bags for resistance); and (6) triceps strength, by pressing on the armrest or using Thera-bands for resistance, and progressing to a wall press.

The intensity of activities progressed gradually by increasing the number of repetitions and level of difficulty throughout the 12-week intervention. The program was delivered by an experienced instructor, registered as a BC Recreation and Parks Association certified fitness instructor and as a certified instructor for the BC Women’s Hospital Osteofit program. Two research assistants were present during each session to facilitate program delivery. The research assistants documented each participant’s attendance and amount of exercise completion in each session as measures of adherence.

### 2.6. Quantitative Outcome Measures

Fall risk and cognitive function: We assessed fall risk using the Longitudinal Aging Study Amsterdam (LASA) fall risk profile questionnaire [16], which has been validated to predict falls [17]. The LASA fall risk profile consists of nine items including fall history, education level, dizziness, body weight, hand grip strength, functional limitations, having a pet, alcohol intake, and concern about falling using Falls Efficacy Scale-International (FES-I) [18]. The total LASA fall risk profile score ranges from 0 to 30; higher scores indicate a higher risk of falls. We used the Montreal Cognitive Assessment (MoCA) to assess cognitive function [19]. The total MoCA score ranges from 0 to 30; higher scores indicate better cognition.

Strength and mobility: We assessed isometric hand grip strength using a Hydraulic Hand Dynamometer (Fabrication Enterprises, White Plains, NY, USA). Participants performed two trials with the dominant hand, with a 30-s rest period between trials. We used the maximum grip-strength (kg) for analysis. To characterize quadriceps strength, we measured knee extension strength of the dominant leg (dominance based on self-report) using a strap with a strain gauge in series [11,20]. Participants sat erect in a chair with their arms crossed and without back support. For statistical analysis, we normalized knee extension strength to the distance between the strap and the knee, to compensate for differences in individual anthropometry.

We assessed functional mobility using the Timed Up & Go Test [21], the Short Physical Performance Battery (SPPB) test [22], and usual walking speed. The Timed Up & Go requires participant to get up from a chair, walk three meters, turn, walk back, and sit back on the chair at their preferred speed, and time taken is registered with a stopwatch. SPPB comprises 5-time sit-to-stand, standing balance, and gait speed assessments over four meters. Scores range from 0 (worst performance) to 12 (best performance). We also used a 10-m timed walk to measure walking speed. Participants were instructed to walk at a usual pace in a straight line. We performed the trial twice. Walking speed (meters per second) was calculated for the middle 8 m of the course to exclude initiation and termination of gait; the mean of two trials was used for analysis.

### 2.7. Statistical Analysis

We used SPSS Statistics version 24 (IBM Inc., Armonk, NY, USA) for statistical analyses. We compared baseline characteristics of the groups using *t*-test and Fisher’s Exact Test. We described the mean and 95% confidence interval of outcome variables as well as percent of change from T0 to T1 within each group. For the main analysis, we used non-parametric statistics to assess effects of Mobility-Fit compared to Usual Care, since data were not normally distributed. Specifically, we compared changes in outcome measures (from T0 to T1) between the Mobility-Fit group and Usual Care groups using Mann Whitney U tests. We set the two-sided level of significance at α = 0.05. Effect sizes were expressed as a Cohen’s *f* and interpreted as small (*f* = 0.10), moderate (*f* = 0.25), and large (*f* = 0.40) [23].

### 2.8. Qualitative (Implementation) Outcome Measures

Program delivery: At completion of the 12-week intervention, we collected participants’ perspectives on the Mobility-Fit program using an evaluation survey form, which is a standard evaluation form provided by Fraser Health Authority. Nine (82%) participants in the Mobility-Fit group completed the survey. We also conducted semi-structured interviews to identify barriers and facilitators to delivery of the Mobility-Fit program with four participants (2 men and 2 women) and three care staff (1 manager and 2 recreation practitioners). Questions directed to staff focused on program delivery, i.e., perceived success and challenges.

Perceived effectiveness: We evaluated participant’s perception of the interest, benefit, intensity of the program, and behaviour change, as measures of perceived effectiveness. Each interview was conducted and audio-recorded by a team member, which took about 30 min to complete. We then undertook thematic analysis [24]; three researchers generated and summarized common themes of the interview data.

## 3. Results

### 3.1. Participants Characteristics

We recruited residents from two AL facilities; 20 (18% of residents) from ELIM, and 10 (16% of residents) from SC. All 30 participants completed baseline assessments; eight withdrew from the study before completion (four in each group) (Figure 1). Among the four participants who withdrew from the Mobility-Fit group, two were hospitalized for acute conditions and two (a couple) preferred not to continue. Of the four who withdrew from the Usual Care group, two were hospitalized for surgery, one had an acute condition, and one chose not to participate in the final (T1) assessment.

Twenty-two participants completed T0 and T1 assessments, with 11 in each group. We provide baseline characteristics of all participants in Table 1. They were 85 (SD 4.5; Range 65–95) years; 67% were women; and had a MoCA score of 22 (SD 5) indicating mild cognitive impairment, and 53% used a mobility aid. The average length of stay in AL was 2.4 (SD 2) years. Participants in the Mobility-Fit group had lower body mass index than those in the Usual Care group (*p* = 0.020, based on *t*-test). Age, sex, length of stay, use of mobility aids, and having fall(s) in the past 12 months did not differ significantly between Mobility-Fit and Usual Care groups at baseline.

### 3.2. Quantitative Outcomes

We report means (95% confidence intervals) and changes in outcome measures from baseline (T0) to completion (T1) for participants within the Mobility-Fit and within the Usual Care group (Table 2). We also compare changes in outcome measures (from T0 to T1) between Mobility-Fit and Usual Care groups. Participants in the Mobility-Fit group increased knee extension strength while the usual care group had decreased strength (*p* = 0.031). Dominant hand grip decreased in both groups; however, the decrease in the usual care group was significantly greater (*p* = 0.047). Further, the Mobility-Fit group improved five-time sit-to-stand and reaction time while the usual care group’s performance for these tests decreased (*p* = 0.034 and *p* = 0.016, respectively). Furthermore, the Mobility-Fit group also showed a tendency of improved LASA fall risk scores while usual care group performance decreased (*p* = 0.056). Effect sizes were large for MoCA, hand grip strength, knee extension strength, SPPB, and walking speed; we noted medium effect sizes for all other outcomes.

### 3.3. Implementation Outcomes

Program delivery: Among Mobility-Fit participants, the adherence rate to the sessions was 77%. Four participants attended all sessions. Five participants completed approximately 70% and two completed approximately 50% of sessions. As a measure of fidelity, most participants completed 80 to 100% of activities in each session. Reasons for not completing an activity included “lack of motivation”, “tired”, and “unable to perform the activity”.

Perceived effectiveness: The perspective of participants regarding the Mobility-Fit program was generally positive. Approximately, 67% rated the program as “very good” (Table 3); 56% felt the program benefited them “very much”. Among participants, 44% expressed they would “most likely” change their lifestyle to become more active. When asked if they would participate in the program again, 56% answered “definitely yes”. From their comments, participants noted: “Felt more confident”, “Felt more awareness of the need to keep active”, “I can move around a little better”, and “I learn that even a light exercise, do it the right way consistently, can make a huge difference”.

Interviews probed more deeply into the perceived benefits and challenges encountered by residents who participated in Mobility-Fit; several themes emerged. Participants enjoyed the program very much. They liked all the activities they engaged in during the delivery session, particularly those that focused on improving balance. Two male participants who had never joined any previous physical activity programs in the facility indicated, “I liked the change of different activities in the class and progression of the program as it created challenges”, and “The real-time feedback by the instructor is encouraging”. One man and one woman expressed that “hand pushing against the wall was a good exercise to strengthen the arms but was a bit difficult”. Participants also continued physical activity on their own at home or at the gym, after participating in Mobility-Fit. One participant commented that “the intensity and duration of the program were appropriate”—a sentiment shared by all other participants. Another participant indicated, “I would like to attend the program again if offered. I like the class to have a variety of activities”.

Staff members commented that “the program was successfully delivered, and participants liked it”. However, resident’s enthusiasm to participation in Mobility-Fit was slightly lower than for other programs offered in the facility, such as dancing or social outings. Staff noted that “it was not a high participation rate, as they had conflict with other routines” and “they did need to adapt to the new program with new instructor”. Staff suggested that repeatedly highlighting the benefits of Mobility-Fit to participants during program delivery and providing sample sessions ahead of enrolment for Mobility-Fit would improve participation. Staff also suggested that using technology to deliver exergames (e.g., video games, Wii, Xbox) may motivate AL resident participation.

## 4. Discussion

We extend the literature by adapting an evidence-based program [11] to create Mobility-Fit, a program that aims to address fall risk factors in older adults who reside in AL. Our Mobility-Fit efforts were motivated by our previous studies in residential care that demonstrated that chair transfer due to poor arm strength was a common cause of falls [9], and that hand arrest was not effective in preventing head impact [8,10]. Thus, a very purposeful and unique component of Mobility-Fit was the emphasis on strengthening the upper-limb and core to enhance safe mobility (e.g., transfer to and from chairs and to stabilize posture through hand support), and to arrest falls to reduce the risk of injury when falls occur. Mobility-Fit results were promising and extended targeted activity choices for older adults and care providers in the AL context, where few fall prevention studies have been conducted [5]. Furthermore, we used qualitative methods to assess two aspects of implementation from the perspective of residents and program staff: program delivery and perceived program effectiveness. This provided us the novel opportunity to delve into factors that may explain outcomes we observed at the participant level.

### 4.1. Effects of the Mobility-Fit

Consistent with our hypothesis, Mobility-Fit participants improved or maintained their muscular strength, while strength of participants in the Usual Care group continued to decline. Improved lower limb strength (knee extension) in the Mobility-Fit group aligns with findings from a previous Osteofit study [11]. Osteofit was specifically designed to enhance lower limb strength and prevent falls in older women diagnosed with osteoporosis. We retained the lower limb component while adding activities that promoted upper limb strength. Despite this, upper limb strength did not improve in the Mobility-Fit group. It has to be noted that we used hand grip strength, which may not be very amenable to change through exercise, as indicator of upper limb strength. However, and likely as important, grip strength was maintained in the Mobility-Fit group but declined (6–9%) in the Usual Care group. A loss of muscular strength reflects greater frailty of older adults in AL, a persistent by-product of high levels of sedentary behaviour [3]. Across 10 countries, older adults were sedentary for nearly 10 hours per day [25]. Low levels of activity are exacerbated in older adults who reside in AL facilities, most of whom spend 86% of waking time being sedentary [3]. Participant’s mobility limitations may have compromised their ability to progressively increase activities to an intensity required to prompt significant changes in some outcomes. Alternatively, activities may simply not have been intense enough or of sufficient duration to stimulate changes. In the Staying on Your Feet program [12], training on upper limb strength was the focus (>50%) of the program, and hand grip strength and elbow extensor strength were improved among community-dwelling older women. However, in our program, upper limb strength activities comprised only 15–20% of the Mobility-Fit session. Our preliminary results suggest that a longer duration may be needed to generate significant improvement in upper limb strength, in future studies.

Exercise programs with agility and coordination components improved reaction time in older adults [26,27,28]. Faster reaction times in the Mobility-Fit group may be associated with targeted agility and coordination training. While central processing time may be compromised among inactive older adults, it can be stimulated and enhanced through physical activity [26,28]. For an upper limb protective response to be effective, the person must quickly and accurately move the upper limbs into a position for fall arrest and generate sufficient muscle forces to arrest the motion of the torso and head [29]. However, falls occur over a brief time scale; head impact may occur in less than a second after descent is initiated [30]. This limits the time available for a faller to accurately position the upper limbs to recover from balance loss or to arrest a fall. The inability to execute a protective response is exacerbated in more frail older people. Thus, improved upper limb reaction time may be key to successfully arresting a fall. Further research may explore relationships between reaction time and effectively arresting a fall using the upper limbs in older adults.

The tendency of decline in the LASA fall risk scale in the Usual Care group may be explained by changes in hand grip strength and fall efficacy—two major components of the LASA score. The Mobility-Fit group improved fall efficacy score by 28%. However, given the small sample size the change of LASA score was not statistically significant. Importantly, by the end of the study, participants in the Mobility-Fit group felt more confident performing daily activities. Thus, Mobility-Fit may be an attractive choice to promote physical activity self-efficacy in AL.

The observation of improved or maintained mobility in the Mobility-Fit and decline in Usual Care groups was driven by the difference in participant’s sit-to-stand times, which was 15% greater improvement in the Mobility-Fit group, but 18% decrease in the usual care group. Improved chair transfer is likely a function of greater knee extension strength in the Mobility-Fit group, consistent with findings in other studies [11,31]. Meanwhile, the increased SPPB score (0.3) in the Mobility-Fit group is considered clinically relevant for sedentary older adults. Specifically, a change of SPPB score between 0.3–0.8 points is considered as minimally significant change in exercise interventions [32].

### 4.2. Feasibility of Delivery of the Mobility-Fit

It remains a challenge to engage and retain a high proportion of older adults in physical activity in the AL setting. Attrition in our study was 27% (8/30), which is similar to other studies conducted in AL [33,34,35] but greater than in studies of community-dwelling older adults [11,12]. This result also reflects a decline of physical function and more illness in AL residents generally [3], and in our participants specifically (5/8 participants discontinued due to illness). Adherence (number of sessions attended) to Mobility-Fit sessions (77%) was similar to adherence in studies conducted in institutional settings [28,36]. Low adherence rate in older adults in institutions may be due to mobility limitations and low self-efficacy in this population as they are more susceptible to illness [28,36].

Importantly, Mobility-Fit was delivered with fidelity in collaboration with the Health Authority with whom the facilities were affiliated. Fidelity to the program (80–100% of the activities performed as planned in each session) reflects its appropriateness and progressive design. All participants began Mobility-Fit activities at low intensity and progressed to relatively higher intensity activities across 12 weeks. There were no major adverse events although two participants experienced sore muscles and feelings of exertion. Mobility-Fit was well-supervised by an instructor and two facilitators; thus, fidelity to the program and positive perceptions about its delivery and effectiveness may have resulted from this personalized approach. A personalized approach is particularly effective for older adults with physical or cognitive impairments [5]. Our results suggest that Mobility-Fit is a feasible program for frail older adults in AL.

Several other factors likely facilitated effective delivery and perceived effectiveness of Mobility-Fit. First, instructors provided clear instruction and demonstrated all activities. Real-time feedback from instructors regarding correct performance of an activity enhanced the performance and self-efficacy [37]. Second, Mobility-Fit required minimal equipment (e.g., Thera-bands), which made program delivery more feasible and flexible. To illustrate, each participant chose a resistance band based on their physical capacity. Further, activities that comprised the program could also be adapted for each person’s physical capacity. Third, the progressive nature made the program more appropriate, more doable, and more engaging for this cohort. Participants increased the level of difficulty based on their physical capacity and personal progress. It is important to note that two male participants never attended exercise programs in the facility previously as they felt activities were too simple; however, they fairly liked Mobility-Fit’s emphasis on strength with progression.

We encountered several barriers to delivery of Mobility-Fit. First, scheduling conflicts may have decreased individual participation in Mobility-Fit sessions. Competing activities within AL facilities were social outings, knitting class, and bible reading. Second, illness affected participation; five participants withdrew due to illness. This is common for older adults who reside in AL as they often experience declines in health over time [28,36]. Third, other residents disrupted the flow of Mobility-Fit delivery [38]. For example, some residents not enrolled in the study joined the sessions mid-way through its 12-week duration. It was not possible to avoid this as staff encouraged all residents to actively engage in facility programs. This may be an important practical consideration for actually running programs in AL.

### 4.3. Limitations

We acknowledge that our study had several limitations. First, participants were physically independent, as per our inclusion criteria. Thus, those who were healthier and more motivated than their peers in AL volunteered to participate. This may have introduced sampling, response, and performance bias. However, as Mobility-Fit is flexible and adaptable to physical capacity, it could be delivered to a frailer population in the future. Second, the sample size is small, and we used a quasi-experimental design; thus, participants were not randomized, nor was the research team blinded to group assignment. This may have led to disparity between groups at baseline. However, the main purpose of this study was to evaluate the feasibility of the novel Mobility-Fit program. Results from this study will guide the sample size and power calculation for a randomized controlled trial, which will be based on a mixed-design ANOVA with random assignment and covariates adjusted (e.g., age, health conditions), to further assess the effect of Mobility-Fit for older adults in care facilities. Third, we experienced 27% drop-out (eight out of 30). However, five drop-outs were due to illness, which is difficult to predict and prevent, especially in AL where many older adults have chronic conditions. Despite this, our findings hold promise and can be used to design a larger study with greater power to detect differences in the future. Fourth, the age range is wide, which may have affected our results. However, this is the nature of residents in assisted living where admission depends on older adults’ functional capacity but not merely on age. Nevertheless, despite the wide age range in our sample, functional capacities of most participants are similar. Finally, only two men completed Mobility-Fit. However, the gender distribution in our study is similar to that in assisted living in Canada where over 70% of residents are women. Our study suggests the feasibility of Mobility-Fit for both men and women, but also highlights the need for gendered approaches to interventions and implementation strategies that engage older men in physical activity programs.

## 5. Conclusions

The general trend is that older adults in AL become increasingly frail as their physical and cognitive abilities decline. Thus, global strategies that effectively promote mobility and offset physical decline are sorely needed. Results from our study suggest that well-conceived, progressive programs that specifically target fall risk factors such as diminished lower limb strength, reaction time, upper limb strength and mobility function may be one solution to enhance mobility in this frail population. Importantly, implementation may be enhanced through a personalized approach, real-time feedback regarding physical activity performance, and program facilitators who assist to promote effective program delivery. Our findings can be used to guide adaptation and implementation of Mobility-Fit at a broad scale, to preserve the function and mobility of frail older adults.

## Figures and Tables

**Figure 1 ijerph-19-05453-f001:**
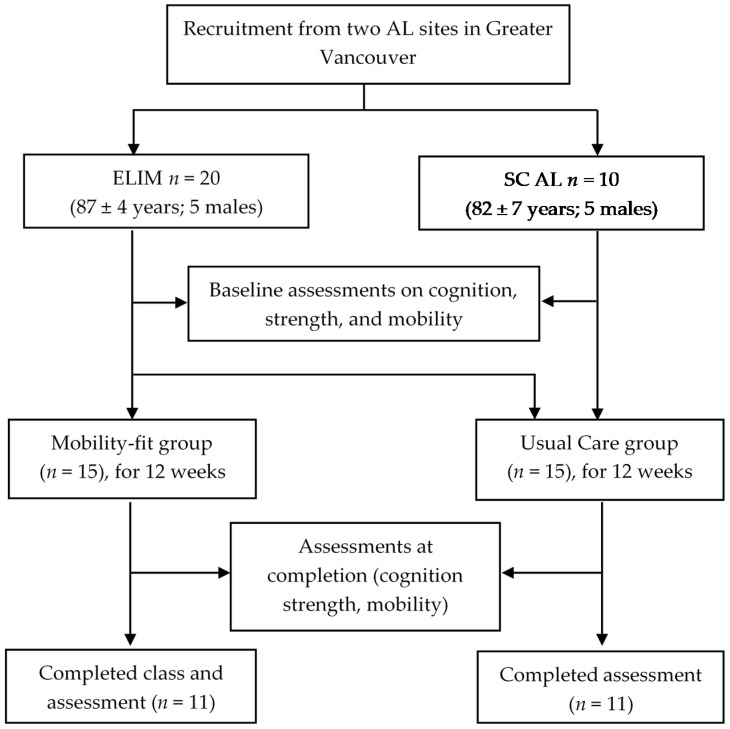
Flowchart of group allocations for the Mobility-Fit and Usual Care groups in this study.

**Table 1 ijerph-19-05453-t001:** Baseline characteristics of participants in Mobility-Fit and Usual Care groups.

Baseline Characteristics	Mobility-Fit (*n* = 15)	Usual Care (*n* = 15)	*p*-Value
Age (years; mean, SD)	87 (5.1)	84 (6.7)	0.084
Female (*n*, %)	11 (73.3%)	9 (60.0%)	0.700
Length of stay (years; mean, SD)	3.1 (3.0)	1.8 (1.1)	0.132
Use of mobility aids (*n*, %)	6 (40.0%)	10 (66.7%)	0.272
Body mass index (mean, SD)	23.2 (3.1)	27.9 (5.8)	0.020
Self-reported having fall(s) in past year (*n*, %)	5 (33.3%)	9 (60.0%)	0.272

Note: *p*-values are based on *t*-test (for continuous variables) and Fisher’s Exact Test (for categorical variables).

**Table 2 ijerph-19-05453-t002:** Outcome measures bootstrapped mean (bias-corrected and accelerated 95% confidence interval) for Mobility-Fit and Usual Care groups at baseline (T0) and after 12 weeks (T1).

	Mobility-Fit (*n* = 11)			Usual Care (*n* = 11)			Comparison		
	Pre-Test (T0)	Post-Test (T1)	Change from T0 to T1	Pre-Test (T0)	Post-Test (T1)	Change from T0 to T1	Mann Whitney *U*	Common Language Effect Size *f*	*p*-Value
**Fall risk and cognitive function**									
LASA fall risk score	4.2 (2.8–5.7)	3.6 (2.2–4.9)	−14%	4.7 (2.4–8.0)	5.2 (3.4–7.1)	11%	31.00	0.26 ^m^	0.056
FES-I score	5.2 (3.2–7.3)	3.8 (2.6–5.3)	−28%	9.3 (7.6–11.1)	9.4 (8.1–11.2)	1%	33.00	0.27 ^m^	0.076
MoCA score	21.3 (10.8–25.1)	22.4 (19.4–25.3)	5%	21.7 (20.1–23.3)	22.1 (20.7–23.6)	2%	75.00	0.62 ^l^	0.365
**Strength**									
Dominant hand grip (kg) *	20.9 (17.2–26.3)	20.9 (17.9–24.7)	0%	21.9 (16.2–28.9)	20.6 (14.3–28.0)	−6%	91.00	0.75 ^l^	0.047
Knee extension strength (kg) *^,#^	20.9 (18.0–23.8)	22.2 (18.9–25.6)	6%	23.4 (18.9–28.6)	21.5 (16.6–27.4)	−8%	78.00	0.79 ^l^	0.031
**Mobility**									
Timed Up and Go (s)	12.9 (11.0–15.4)	13.0 (10.9–15.6)	1%	20.6 (16.9–24.1)	23.0 (17.6–28.8)	12%	46.00	0.38 ^m^	0.365
SPPB score	8.3 (6.9–9.5)	8.6 (7.6–9.4)	4%	5.0 (4.2–5.8)	5.0 (4.3–5.7)	0%	70.50	0.58 ^l^	0.519
5 time sit-to-stand (s) *	19.8 (16.6–23.3)	16.9 (15.3–18.6)	−15%	21.4 (18.5–24.6)	25.2 (20.1–30.4)	18%	28.00	0.23 ^m^	0.034
10-m gait speed (m/s)	1.08 (0.90–1.24)	1.11 (0.94–1.27)	3%	0.73 (0.62–0.85)	0.75 (0.64–0.87)	3%	59.00	0.49 ^l^	0.949
Reaction time (s) *	0.38 (0.30–0.46)	0.32 (0.26–0.39)	−16%	0.31 (0.25–0.36)	0.37 (0.29–0.48)	19%	24.00	0.20 ^m^	0.016

Note: * *p* ≤ 05. Results were based on Mann–Whitney U test for the comparison between groups; ^#^ Data were missing for two usual care group participants; ^m^, medium Cohen’s f effect size; ^l^, large Cohen’s f effect size.

**Table 3 ijerph-19-05453-t003:** Results from the program evaluation form completed by participants in the Mobility-Fit group (*n* = 9).

Question	Rating by Participants				
How did you like the program?	Excellent (0%)	Very good (66.7%)	Good (22.2%)	Average (0%)	Fair (11.1%)
Did the program benefit you?	Extremely (0%)	Very much (55.6%)	Somewhat (44.4%)	Minimally (0%)	No benefit (0%)
How often did you participate?	All the time (11.1%)	Most of the time (88.9%)	Sometimes (0%)	Barely (0%)	Never (0%)
Timing of the class?	Very convenient (11.1%)	Convenient (66.7%)	Somewhat (22.2%)	Not convenient (0%)	
Any lifestyle changes?	Definitely yes (0%)	Most likely (44.4%)	Maybe (44.4%)	Most likely not (11.1%)	Definitely not (0%)
Would you participate again?	Definitely yes (55.6%)	Most likely (33.3%)	Maybe (11.1%)	Most likely not (0%)	Definitely not (0%)

## Data Availability

The datasets used and analyzed in the study are available from the corresponding author on reasonable request.

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
