# Peer review of "Effects of the Mobility-Fit Physical Activity Program on Strength and Mobility in Older Adults in Assisted Living: A Feasibility Study"

_ijerph, 2022, doi:10.3390/ijerph19095453_

Round 1

Reviewer 1 Report

In my opinion, the investigation is sufficiently justified.
It has several limitations, which are listed in the corresponding section of the discussion. From my point of view, it also has these two limitations: a) the sample is very small, b) the age range is very wide SD 4.5. At these ages this aspect is very relevant. For all this, I suggest the authors include them in the corresponding section.

Author Response

We thank the reviewer's positive feedback and agree with the comments. We have added the small sample size and wide age range to the Limitations. New text that has been added as part of this revision is indicated in underlined text below. 

Lines 421-444. Limitations

We acknowledge that our study had several limitations. First, participants were physically independent, as per our inclusion criteria. Thus, those who were healthier and more motivated than their peers in AL volunteered to participate. This may have introduced sampling, response and performance bias. However, as Mobility-Fit is flexible and adaptable to physical capacity, it could be delivered to a frailer population in the future. Second, the sample size is small and we used a quasi-experimental design; thus, participants were not randomized, nor was the research team blinded, to group assignment. This may have led to disparity between groups at baseline. However, the main purpose of this study was to evaluate the feasibility of the novel Mobility-Fit program. Future randomized controlled trials with larger sample sizes should definitively assess the effect of Mobility-Fit for older adults in care facilities. Third, we experienced 27% drop-out (8 out of 30). However, 5 drop-outs were due to illness, which is difficult to predict and prevent, especially in AL where many older adults have chronic conditions. Despite this, our findings hold promise and can be used to design a larger study with greater power to detect differences in the future. Fourth, the age range is wide, which may have affected our results. However, this is the nature of residents in assisted living where admission depends on older adults’ functional capacity but not merely on age. Nevertheless, despite the wide age range in our sample, functional capacities of most participants are similar. Finally, only two men completed Mobility-Fit. However, the gender distribution in our study is similar to that in assisted living in Canada where over 70% of residents are women. Our study suggests the feasibility of Mobility-Fit for both men and women, but also highlights the need for gendered approaches to interventions and implementation strategies that engage older men in physical activity programs.

Reviewer 2 Report

This manuscript has been revised considerably after initial review. The primary concern regarding this manuscript was power and related statistical tests. The authors have effectively troubleshoot this concern by labeling this work as "pilot". I have no revisions to add at this time. 

Author Response

We thank the reviewers for their positive feedback on our revision.

This manuscript is a resubmission of an earlier submission. The following is a list of the peer review reports and author responses from that submission.

Round 1

Reviewer 1 Report

The improvement of physical condition in the elderly has a direct impact on their health and quality of life, therefore, this study seems very appropriate, since it presents evidence on the importance of physical exercise. But it has a very large limitation, and it is the size and heterogeneity of the sample, gender is not compensated, and this is a variable that biases the study. In addition, gender is a determining factor in the evaluation values ​​of physical exercise.
On the other hand, the methodology used is correct, but it presents an important limitation, which is the size of the sample. The sample is very small. On the other hand, it is from a long time ago, specifically from the year 2017. Although this does not detract from the validity of the results.
I suggest that the authors make the following modifications in order to improve the paper.
Lines 71-78. I consider that the objective of the study is not correctly written. It should be stated in one or two lines. The rest of the text is superfluous. I suggest the authors delete the last lines where they talk about the practical implications. The practical implications should be indicated at the end of the article.

Participants: Authors must report the age range of the participants.

Lines 457-472. The authors must analyze whether the age of the participants may be a limitation, the standard deviation is almost seven years. This aspect is decisive to analyze the results, since at these ages the physical condition varies a lot.

I must recommend rejecting the paper, the sample has a very limited representation, with aspects that may present very important biases, such as: the age range of the participants or gender.
On the other hand, it is observed that the authors have extensive research experience and the article is correctly written, and in my opinion, the statistical analysis is adequate. But that the formal aspects are impeccable does not mean that the results are relevant. I insist, the sample is very uneven and too small.

Reviewer 2 Report

IJERPH-1330300 presents results from an intervention in AL facilities. While some parts of this manuscript were interesting, other areas could be improved. I hope the authors consider my feedback.

MAJOR COMMENTS

  • Table 1 and Methods: With a smaller n= it realistically is challenging to suggest the characteristics of the two groups were not statistically different, regardless of the power analysis presented. Be careful with language throughout.
  • Fall Risk Questionnaire: It is possible that overlap is occurring for the LASA and handgrip strength given that handgrip strength is part of the LASA. Is there strong rationale to separate these measures but not others?
  • Lines 174-179: The measure of leg extension strength is not really valid and rarely used in the literature. Expensive equipment is necessary for collecting these data. Consider outright deleting this measure from the paper.
  • Lines 181-188: It needs to be clearer about which measures were included as part of the SPPB, and which measures were examined independently. Also, why were some parts of the SPPB explained (gait) more than others?
  • Intervention Contents: How did other factors such as diet factor into the study?
  • Implementation Outcomes: The adherence of the program is relatively low. This has implications on the findings.
  • Discussion: Consider discussing the negative findings from the STRIDE study (https://www.nejm.org/doi/full/10.1056/NEJMoa2002183). How does this intervention differ from STRIDE and why should it be further pursued?

MINOR COMMENTS

  • Line 40: Consider just replacing “more and more” with “increased” or similar for language truncation.
  • Study Design and Recruitment: How were inclusion and exclusion criteria determined? Self-report?
  • Line 110: Be careful about presenting results (p<0.05) in the Methods without first describing the statistical test. Just seems out of place.
  • Statistical Analysis: Did the authors consider examining measures between the 12-week study period?
  • Qualitative outcome measures: Are there feasibility guidelines that help in the construction of the evaluation survey? This may help with validation.
  • Line 221: Avoid re-introducing figures. Same for Table 1 in line 228.
  • Results: Include the mean and SD differences between the time periods to couple with the p-values in the text.
  • Line 396: Avoid presenting results in a Discussion (e.g., p=0.07).
  • Make any changes to the abstract that align with those made in the text.
